# ESR and Radiocarbon Dating of Gut Strings from Early Plucked Instruments

**DOI:** 10.3390/mps3010013

**Published:** 2020-01-28

**Authors:** Sumiko Tsukamoto, Taro Takeuchi, Atsushi Tani, Yosuke Miyairi, Yusuke Yokoyama

**Affiliations:** 1Leibniz Institute for Applied Geophysics, Stilleweg 2, 30655 Hannover, Germany; 2The consortium for Guitar Research, Sidney Sussex College, Cambridge CB2 3HU, UK; guitarouk@aol.com; 3Graduate School of Human Development and Environment, Kobe University, Kobe 657-8501, Japan; tani@carp.kobe-u.ac.jp; 4Atmosphere and Ocean Research Institute, The University of Tokyo, Chiba 277-8564, Japan; miyairi@aori.u-tokyo.ac.jp (Y.M.); yokoyama@aori.u-tokyo.ac.jp (Y.Y.)

**Keywords:** ESR, gut strings, Fe(III), early plucked instruments, radiocarbon

## Abstract

Early European plucked instruments have recently experienced a great revival, but a few aspects remain unknown (e.g., the gauge of gut strings). Here we report, for the first time, that the electron spin resonance (ESR) signal intensity of oxidized iron, Fe(III), from gut strings at *g* = 2 increases linearly with age within a few hundred years. The signal increase in the remaining old strings on early instruments can be used to judge if they are as old as or younger than the instrument. Obtaining the authenticity information of gut strings contributes to the revival of the old instruments and the music.

## 1. Introduction

Electron spin resonance (ESR) has been utilized as a geochronometer, based upon the increase in the number of trapped electrons and holes in crystal lattices induced by natural radiation with time [1,2,3]. ESR also detects unpaired electrons in organic radicals and transition metals in organic substances; the intensity of such ESR signals also increases with time, mainly by thermal activation processes. The possibility of dating organic materials using organic radicals was first tested using potato crisps [4]. The day-by-day increase in the organic radical intensity was used to estimate the production date of the potato crisps. ESR signals of organic radicals and Fe(III) in organic matters, e.g., animal skins, papers, silks and mummies were also investigated, and the intensity showed a positive correlation with age [5,6,7,8]. Furthermore, Fe(II) in heme-proteins in human blood starts to oxidize to Fe(III) after exposure to air, and therefore the increase in Fe(III) in bloodstains detected by ESR has been suggested for use in forensic investigations [9].

In this study, we test the potential for establishing a relative chronology of gut strings from early plucked instruments using the ESR signals of Fe(III) and organic radicals. Gut strings are made from sheep guts containing heme-proteins with Fe(II), which is hardly detected with ESR. Once Fe(II) changes to Fe(III) by the catalytic cycle reaction of the heme-proteins or oxidation reaction, it becomes detectable with ESR. The *g*-values of the Fe(III) signals in oxidized heme-proteins are affected by the ligand fields [10,11,12]; for instance, the Fe(III) signal at *g* = 6 (high spin state, *S* = 5/2) is originated from methemoglobin coordinated with H_2_O, whereas the signal at *g* = 2 (low spin state, *S* = 1/2) is that coordinated to CN^−^ or OH^−^ [12,13].

Early European plucked string instruments (e.g., lutes, early guitars and harp-lutes) had fallen out of use by the middle of the 19th century, but have been revived with the increased interest in early music in the 20th century. Old gut stings are found occasionally on the instruments or in the cases. We examine five old strings from guitars and harp-lutes, which were made in the 19th century, as well as four strings with known ages. Radiocarbon dating of the strings is also conducted for six strings for a comparison. Although radiocarbon dating has been frequently applied to archaeological music instruments [14,15], little work has been done on recent plucked instruments [16]. To our knowledge, this is the first study of radiocarbon dating on modern (19th–20th centuries) gut strings. 

## 2. Samples and Results

The nine gut strings used for the ESR investigation are listed in Table 1. The approximate ages of four gut string samples are known. Two of them (LHE-14 and -27) were obtained shortly before the first measurement of the samples in April 2015. Another string (LHE-15) was acquired around the year 2000. LHE-16 was provided with a memorandum of the supplier, G. Butler and Sons in London, in 1913 (Figure 1a). Five string samples were collected from four original instruments. Two harp-lutes, which are both assumed to have been made ca. 1815, had remaining old gut strings at the bridge. Two string samples (LHE-17A, -17B) were collected from the 14-string harp-lutes (Figure 1b), and a sample LHE-28 was taken from the 12-string instrument (Figure 1c). A separate string from the 14-string harp-lute (LHE-17C) was used for radiocarbon dating. Sample LHE-19 was found in the original case of a guitar, which was made approximately in ca. 1840 by D&A Roundhloff in London, and LHE-44 was obtained from a guitar, estimated to have been made between 1850 and 1860. All these original instruments were produced and preserved in the United Kingdom. 

ESR spectra of a recent (LHE-15) and an old string (LHE-17A) are shown for wider (250 ± 250 mT; Figure 2a) and narrower (324 ± 5 mT; Figure 2b) scanned magnetic fields. All gut string samples showed ESR signals from Fe(III) signal at *g* = 2 (low spin states, *S* = 1/2) (Figure 2a) and organic radicals at around *g* = 2.005 (Figure 2b). The Fe(III) signal at *g* = 6 (high spin states, *S* = 5/2) (Figure 2a) was observed in all samples except LHE-28.

All these strings are assumed to have been produced within ~2 years before the time of acquisition. Crude but supporting evidence was obtained by the increasing intensity of the Fe(III) signal at *g* = 2 within the 11 months for the two modern strings, LHE-14 and LHE-27 (Appendix A). When the intensity is plotted against the time after the first measurement, April 2015, the x-intercept of the fitted straight line points to 15 months and 19 months before the first measurements for LHE-14 and LHE-27, respectively. 

The correlation of the ESR signal intensity with age was first examined using four string samples with known ages (LHE-14, -15, -16, and -27). The results are plotted in Figure 3a. Although the number of known age samples is limited, it is clear that the best correlation between the ESR signal intensity and age is obtained from the Fe(III) signal at *g* = 2 (*r* = 0.95, Spearman’s correlation coefficient). The other two signals showed much less correlation with age, with correlation coefficient of 0.63 for both organic radicals and the Fe(III) at *g* = 6. This indicates that the Fe(III) signal at *g* = 2 increases linearly with time, for at least 100 years, and therefore can be used for the relative age estimation of older gut strings. 

In Figure 3b–d, the three ESR intensities of all samples are plotted against the known string age (filled symbols) or the expected instrument age (open symbols). The results of Fe(III) at *g* = 6 and the organic radicals are highly scattered, suggesting these two signals are not suitable as chronometers. For the Fe(III) at *g* = 2, the intensity of three of the five old string samples is consistent with the extrapolated fitted line of the known age strings within the 1-σ uncertainty (LHE- 17A, -17B, and -28) and one is consistent within the 2- σ uncertainty (LHE-19). This suggests that these strings are as old as the instruments. However, the Fe(III) signal (*g* = 2) of LHE-44 has a much lower intensity when compared to the expected intensity from the age of the instrument. 

The results of the radiocarbon dating are summarized in Table 2. The method successfully distinguished the strings before and after 1950. The three recent samples (LHE-14, -15, -27) yielded > 100% modern carbon (pMC), and are therefore judged as modern. The ^14^C ages of the older three string samples (LHE-16, -17, and -44) ranged from 167 ± 26 y BP to 309 ± 29 y BP. These were calibrated using OxCal v.3.10 based on Intcal13. Detailed results of the calibrated ^14^C ages are given in Appendix A.

## 3. Discussion

The oxidation of Fe(II) to Fe(III) should be sensitive to various environmental factors, e.g., temperature and moisture. As shown in Figure 3a, the very good positive correlation of the Fe(III) signal at ***g*** = 2 is probably due to the fact that all the string samples used in this study were preserved in the United Kingdom, and therefore the environmental factors were similar. A possible explanation of contrasting correlation with age for these two types of Fe(III) is that the low spin heme-proteins are contained relatively uniformly in all gut strings, but the concentration of the high spin heme-proteins is different from sample to sample. 

The observed increase in the Fe(III) signal at *g* = 2 does not show clear tendency toward saturation, although it is natural to assume that the signal intensity reaches saturation over a longer time period, since a limited number of Fe(II) in heme-proteins should be available in the gut strings. The blue dashed line in Figure 3b shows a regression line, fitted to a single saturation exponential function when all Fe(III) data at *g* = 2 except LHE-44 are used. The fitted line is not significantly deviated from the linear regression line (middle of black solid lines in Figure 3b), suggesting that the signal increase is still very close to linear for ~200 years. This result is in contrast with the study of the Fe(III) signal in coagulated human blood; in which the intensity of high spin state Fe(III) reached close to saturation in a few hundred hours at room temperature after coagulation [9], presumably because the Fe(III) in blood was directly exposed to air and the oxidization process was faster. 

To investigate how the Fe(III) intensity at *g* = 2 in gut strings changes over a longer timescale, aging experiments were conducted using LHE-27 by heating at 60 and 70 °C in air. The result is shown in Appendix A. At 60 °C, the signal increased only about 14% from the modern string intensity and reached saturation, whereas at 70 °C the signal intensity once increased slightly then decreased. In nature, the gut strings of original instruments of ~200 years old (LHE-17A, -17B, -19, and -28) showed 12–14 times larger signal intensity than the modern strings. We conclude that the heating at higher temperature cannot reproduce the natural signal growth.

Using the linear increase in the Fe(III) signal (*g* = 2) of gut strings, it is possible to estimate the age ranges of strings with unknown ages. For the five string samples obtained from early plucked string instruments, the consistent ESR signal intensity with the linear regression line of the known age strings for the four samples (LHE-17A, -17B, -19, and -28) indicate that the strings are as old as the instruments. This indicates that these instruments (early guitars and harp lutes) were not used for a long time after they were made. This also suggests that remaining strings can be used as a record of the original gauges. One string sample, LHE-44 yielded much a lower Fe(III) intensity than expected considering the instrument age. By comparing the intensity with the regression line, it is assumed that this string was made 60–70 years before 2015. 

Currently, it is still unknown how variable the signal increase is in different storage conditions. More data of string samples with known ages and from different environments should be accumulated to establish the method to be a robust dating technique.

AMS radiocarbon dating can accurately judge whether a gut string was made before or after 1950 (Table 2). However, calibration of a ^14^C age into a calendar age generates a large uncertainty (Appendix A). Sample LHE-16, which was sold in 1913, yielded a calibrated age of AD1660–1690 (18.2%), AD1730–1810 (53.4%) and later than AD1920 (23.8%) in 95.4% probability. As mentioned above, the ESR of LHE-44 gave an assumption that the string was much younger than the instrument (AD1850–1860). However, the calibrated ^14^C age of this sample yielded AD1660–1700 (16.7%), AD1720–1820 (51.9%), AD1830–1880 (6.5%) and later than AD1910 (20.5 %), which cover both the instrument age and ESR age of this string. For LHE-17, the calibrated ^14^C age yielded a much older age than the instrument (ca. 1815)—AD1480–1650—which might be due to the contamination of old carbon in the production process.

## 4. Methods

All ESR measurements were conducted using a JEOL-FA-100 X-band spectrometer at the Leibniz Institute for Applied Geophysics, Hannover, Germany. The string samples, which were cut into ~1 cm length, were inserted into quartz glass tubes of 4 mm outer diameter (3 mm inner diameter), and used for the ESR measurements. The measurements were made at room temperature, three times in April 2015, January 2016 and February 2016. The measurement parameters for the Fe(III) signals were; 250 ± 250 mT magnetic field, 1 mW microwave power, 1 min scan time for 3 times, 0.5 mT modulation amplitude and 0.1 s time constant. The organic radical signal (***g*** = 2.005) was measured with the following conditions; 324 ± 5 mT magnetic field, 2 mW microwave power, 30 s scan time up to 50 times, 0.3 mT modulation width and 0.1 s time constant. The mean peak to peak ESR intensity of the three measurements and its 1-σ standard error was calculated. For one sample, which we obtained in December 2015 (LHE-44) the mean signal intensity of two measurements was used. Since the available sample mount varied significantly (1.5 and 26 mg; Table 1), all ESR signal intensities were normalized to the weight of each string sample. 

The radiocarbon dating of the six string samples (LHE-14, -15, -16, -17C, -19 and -44) was conducted with an NEC 250 kV single-stage accelerator mass spectrometer (AMS) at the Atmosphere and Ocean Research Institute, The University of Tokyo [17]. The string samples were washed with ultra-pure water and dried. These samples were oxidized using a Vario Micro Cube elemental analyzer manufactured by Elementar Analysensysteme GmbH., and an AMS analytical target was prepared using an automated graphitization process device manufactured by Koshin Rikagaku Seisakusho Co., Ltd (Tokyo, Japan). 

## 5. Conclusions

A very good correlation between the Fe(III) signal and the known ages of gut strings was observed. This probably indicates that Fe(II) in the gutstrings has been oxidized to Fe(III) with age. We conclude that it is possible to assume whether a gut string is as old as the instrument or much younger, using the correlation of the Fe(III) signal (*g* = 2) with age. Using ^14^C dating it is also possible to judge a string is a recent one or older, but a calibrated age has a large uncertainty.

## Figures and Tables

**Figure 1 mps-03-00013-f001:**
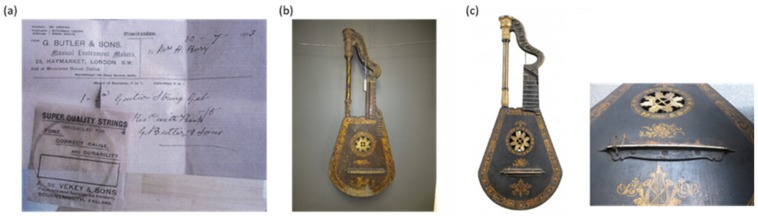
Photos of (**a**) gut string and its receipt (LHE-16), (**b**) a 14-string harp-lute (LHE-17A and -17B) and (**c**) a 12-string harp-lute (LHE-28). Gut string samples were collected from them.

**Figure 2 mps-03-00013-f002:**
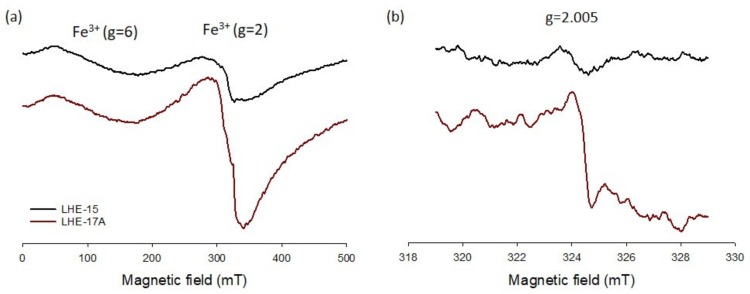
ESR spectra of LHE-15 and -17A for the magnetic field of (**a**) 250 ± 250 mT and (**b**) 324 ± 5 mT. Note that the spectra in (**b**) were recorded with eight times larger receiver gain than (**a**) due to the weak signal intensity.

**Figure 3 mps-03-00013-f003:**
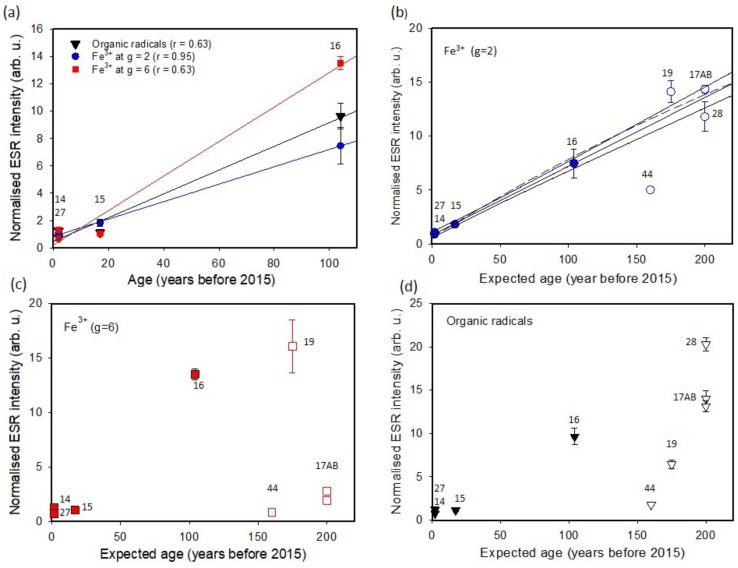
Plot of ESR intensity against the known (filled symbols) or the expected instrument age (open symbols) for (**a**) all ESR signals of the known age samples, (**b**) Fe(III) at *g* = 2, (**c**) Fe(III) signal at *g* = 6, and (**d**) organic radicals. The numbers besides the symbols are sample IDs. All error bars in vertical axis are 1-σ standard error.

**Table 1 mps-03-00013-t001:** List of sample strings for ESR.

Sample ID	Expected Production Year	Weight (mg)	Instrument
LHE-14	ca. 2013	13.6
LHE-15	ca. 2000	15.3
LHE-16	ca. 1913	1.5
LHE-17A	ca. 1815	4.9	14-string harp-lute
LHE-17B	ca. 1815	5.6	14-string harp-lute (same as above)
LHE-19	ca. 1840	1.6	a case of a D&A Roundfloff guitar
LHE-27	ca. 2013	24.8
LHE-28	ca. 1815	26.2	12-string harp-lute
LHE-44	ca. 1850–1860	20.1	from a guitar

**Table 2 mps-03-00013-t002:** Results of radiocarbon dating of the strings.

Lab. Code	Sample ID	δ^13^C (‰)		pMC (%)		^14^C Age (Year BP)	
YAUT-021827	LHE-14	−25.2	± 1.7	104.57	± 0.38	modern
YAUT-021828	LHE-15	−25.4	± 1.8	114.34	± 0.42	modern
YAUT-021830	LHE-16	−30.1	± 0.5	97.73	± 0.27	185	± 22
YAUT-021832	LHE-17C	−24.0	± 1.7	96.23	± 0.35	309	± 29
YAUT-021833	LHE-27	−16.5	± 1.3	118.13	± 0.37	modern
YAUT-021834	LHE-44	−20.4	± 1.3	97.95	± 0.32	167	± 26

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
