# Peer review of "ESR and Radiocarbon Dating of Gut Strings from Early Plucked Instruments"

_mps, 2020, doi:10.3390/mps3010013_

Round 1

Reviewer 1 Report

The authors reported that the ESR signal intensity at g=2 of Fe3+ which contained in gut strings increases linearly with age, and concluded that the signal increase in remaining old strings on early instruments can be used to judge if they are as old as the instrument or younger.

The experimental results obtained show that the ESR signal intensity increase linearly with the gut strings age. However, the mechanism of the signal increase is not clear.

The ageing of hem iron in gut strings may be influenced by storage condition such as temperature and humidity.

This paper will be much better if the basic study of ESR signal for hem iron in dried substance is performed.

There are some minor comments as follows.

Minor comments

Table 1

LHE-17AB → LHE-17A, -17B

3a

The figure is not clear. The sample IDs should be added.

Line 96-97

Further interpretation should be added.

Reviewer 2 Report

The authors used classical EPR methods to determine the relative age of (sheep) gut strings from early plucked instruments. Assumingly this is based on the oxidation of Fe(2+) to Fe(3+) which is found in proteins of sheep guts. The oxidation process is a function of time, hence the signal of Fe(3+) is expected to increase as a function of time. However, I see multiple critical points in the manuscript, and the proposed method in general.

Major remarks/questions/issues:

If I understand correctly: the age determination is based on the EPR signal intensity which is a measure of Fe(3+) concentration. The Fe(3+) is the result of Fe(2+) oxidation. However, oxidation, in general, is strongly dependent on the environment. If the string was used\stored in a strongly oxidizing environment, then the Fe(3+) will increase stronger compared to a neutral environment. How can the method determine the age of string, if the process is strongly dependent on the environment? In table 2. I don’t understand how the ages of sample LHE-16 and LHE-17C as well as LHE-44 are determined as being produced hundreds of years in the future. In particular as the error is given as about 30 years. The measurement may give false information, which can happen, but then stated error is an order of magnitude off. Why all the fits are linear as a function of time? That means that one can measure a sample, which is 10 thousand of years old, which would deliver a huge EPR signal. This is counter-intuitive, as at some point all Fe(2+) is converted to Fe(3+). I would rather expect a non-linear behavior, at least for older instruments. Clearly, the oxidation process might be linear in the beginning, but I have no feeling about the time-interval where one could assume a linear behavior. At least some reference should be stated to support the line fit. Based on remark 3: What is the maximum time frame to which this method can be applied. In general, the reference list could be much longer to support the underlying theory, for both EPR and carbon method.

General remarks:

103: I don’t understand the the sentence: “The exact origins of the observed ESR signal are unknown”. The manuscript states that the origin of the ESR signal is Fe3+. Do the authors mean that the process of oxidation is unknown? If so please rephrase this sentence. How was the samples prepared for the EPR spectrometer? Was a powder made, or was the string placed into the EPR spectrometer as it is? Was the amount of sample same for each measurement? Some more sentences would give the reader a better understanding how the EPR experiment was performed, and more important, allow for other researcher to confirm or use this method. 47: “….LHE-16 was provided by James Westbrook.” It is very polite from the authors to state the sample provider. However, a better place to put such information is in the acknowledgements. The EPR spectrum shown in Fig2b). Is the signal from Fig.2 b seen in Fig 2a as well? If yes, is it a zoom in? If not, what is the difference between the measurements? Although I see the different parameters used in the experiments, but I do not understand why the signal is that different.

Minor remarks:       

2 a and b. Please add a y-axis with proper labelling. Please change (arb.) to (arb. u.) In Fig 3. a): (arb. u.) is missing
